# The Coatings Breakdown Products Influence on the Gas Metal Arc Welding Parameters

**Leonid Zhabrev, Dmitry Kurushkin \*** , **Igor Mushnikov** and **Oleg Panchenko**

Laboratory of Lightweight Materials and Structures, Institute of Mechanical Engineering, Materials and Transport, Peter the Great St. Petersburg Polytechnic University, 29 Polytechnicheskaya St., 195251 St. Petersburg, Russia; leozhabrev@spbstu.ru (L.Z.); mushnikov_iv@spbstu.ru (I.M.); panchenko_ov@spbstu.ru (O.P.)

\* Correspondence: kurushkin_dv@spbstu.ru; Tel.: +7-921-347-6323

**Abstract:** The installation and renovation works of steel structures are often performed using gas metal arc welding. Thereby, the welded elements of these structures are frequently protected by a variety of primers and coatings, especially in shipbuilding. Complex nonequilibrium physical and chemical processes occurring under the influence of high temperatures and electric arc discharge, as well as the presence of the products that affect the welding parameters, have a significant impact on the joints' quality. Experimental studies on the coatings' breakdown products influence on the gas metal arc welding parameters were performed with epoxy, alkyd, polyacrylate, polyvinyl butyral primers, epoxy zinc filled, vinyl chloride, vinyl isobutyl, and organosilicate coatings. The peculiarity of welding current waveform parameters was studied using oscillograms processing. It was found that the main coatings breakdown products that influence the current waveform are oxygen and carbon monoxide.

**Keywords:** weldable coatings; coatings breakdown products; current and voltage waveforms; welding oscillogram; gas metal arc welding

## 1. Introduction

For several centuries, steel has remained the most widespread and demanded structural material, and steel corrosion is a global problem for all industries [1–3]. A solution using various anticorrosion protective coatings is widely implemented in practice worldwide. In a number of industries (shipbuilding and bridge building, nuclear and thermal power engineering, production of reinforced concrete products, etc.), the technology of protecting steel parts and structures is adjacent to the need for assembly, installation, repair, and restoration work performed by arc welding [4–6]. On the one hand, using weldable coatings significantly accelerates and simplifies the structural elements preparation for assembly and welding [7]. On the other hand, the coatings breakdown products reduce the electric arc stability and contribute to an increase in spatter and metal porosity, which can cause a weakening of the structure, especially in the presence of tensile stresses [7]. Moreover, volatile breakdown products can harm the welder's health [8,9].

According to their functional tasks and service life, the materials under study can be divided into shop primers and protective coatings. The shop primers are usually applied with a 10–30 μm layer and are used for the anticorrosion protection of workpieces during transportation, storage, and assembly for a 3–9-month period [10]. They can be used as a first layer in a complex metal protection or removed before final painting using one of the following methods: mechanical, abrasive jet, water jet, chemical, or laser [11]. The removal of the primer by any of these methods requires additional financial and time costs, but even in this case, the use of shop primers is more economical than the rust removing. Primers

certification for welding and cutting (flame, plasma, laser) without their complete cleaning is regulated by international and national regulatory documents (for example, in shipbuilding - by the requirements of Russian Maritime Register of Shipping, American Bureau of Shipbuilding, Bureau Veritas, etc.).

Polyacrylate, alkyd, epoxy, and polyvinyl butyral coatings are popular shop primers. Several works [8,12,13] have provided information on the welding feasibility using coatings based on the above-mentioned polymers and fillers: iron oxide, zinc phosphate, zinc in various forms, etc.

Protective coatings are applied in a much thicker layer, and they usually perform a variety of functions at once: adhesion to the substrate, corrosion resistance, covering, decorative, etc. Their service life, depending on the operating conditions, can reach five or more years. EpoxyCoat-019 and EpoxyCoat Mastic contain a wide range of various nature fillers and are used as a primer or topcoat. The first is used to protect steel against corrosion for: hydraulic structures, free and underwater ship boards, underground pipelines, metal structures, including those with traces corrosion; the second–for internal and external tank surfaces operated in contact with water and oil products.

One of the most demanded coatings, which are used, for example, to protect large-sized bridge structures, are zinc-filled coatings with high barrier properties. Zinc-filled coatings working thickness is 40–120 μm, and if necessary, they can be additionally covered with other materials.

The behavior of organosilicate coatings based on polyorganosiloxanes and hydrosilicates during welding is interesting in a view of their high thermal resistance (for a long time period of 300–600 °C, for a short time up to 1200 °C), which significantly exceeds the temperatures of their formation and onset of organosilicon coatings film former destruction [14]. Depending on the operating conditions and the grade, it is recommended to apply with a layer of 100–200 μm. However, in some cases, a layer 50–80 μm thick is applied at the factory, and the final painting is done after the structure installation. Data on the possibility of conducting electric welding without preliminary organosilicate coatings removal was repeatedly presented in the works of major specialist in the field of organosilicate materials, Nikolay P. Kharitonov. No restrictions were imposed on carrying out electric arc welding with coated electrodes, and it was believed that the organosilicate coatings melt at the metal (steel) melting temperature and form a floating slag without forming inclusions in the weld [15].

Electric arc causes pyrolysis of coatings based on hydrocarbon and organosilicon binders and subsequent degradation products evaporation, the main of which are hydrogen, water vapor, carbon dioxide, less often nitrogen and gaseous compounds based on various metal oxides. The pores are formed as a result of a significant decrease in the gases solubility (primarily hydrogen, nitrogen, and carbon monoxide), which are distributed in the liquid welding pool during its solidification. Other gases, including other volatile coating decomposition products (for example, methane, benzene, formaldehyde, cyclosiloxanes in the case of organosilicate coatings, etc.) do not play a significant role in pores formation [16]. The welding pool liquid metal is heated up to 1727–2127 °C on average, while temperatures of about 2900 °C are reached in the electrode metal droplets and in the welding pool front part in the vicinity of the droplet immersion [17]. At these temperatures, the heated metal, interacting with the gases activated by the arc, can absorb them in quantities that significantly exceed the usual solubility in a solid metal [18–20]. The gas solubility decreases with the subsequent liquid metal cooling and a welding pool is supersaturated with gases in the entire volume [21].

The diversity of the polymer-organic base, the fillers, and the pigments chemical composition additionally complicate the interdisciplinary nature of the discussed problem, and the estimation of the technological advantages and disadvantages is individual for each coating brand. Moreover, the technological parameters of the application and coating drying [13], the arc welding parameters, the joint configuration [22], and the welding consumables preparation [23] have a significant impact on the using weldable coatings possibility in a particular technology.

Since the breakdown products have such an impact on the welding quality, it is necessary to use the most suitable set of the welding parameters in gas metal arc welding (GMAW) with any coating. To choose or develop the special welding set of parameters for a particular set of conditions

(like coating breakdown products presence), current and voltage waveforms and their deviation in these conditions should firstly be studied [24–27].

The aim of the present study was to evaluate the influence of the primers and coatings breakdown products on the welding arc current waveform distortion and deviation. The information obtained in the course of this study is primarily necessary for the development of special waveforms and welding jobs for welding with coatings and primers.

## 2. Materials and Methods

The welding samples were low-alloy mild steel sheets with the dimensions of 10 mm × 200 mm × 300 mm, and one 200 mm × 300 mm side was coated. The chemical composition of the used steel sheets is given in Table 1.

**Table 1.** The chemical composition of the steel sheets.

| C | Si | Mn | Ni | S | P | Cr | N | Cu | As | Fe |
|---|---|---|---|---|---|---|---|---|---|---|
| | | | | | wt.% | | | | | |
| 0.14–0.22 | 0.15–0.3 | 0.4–0.65 | ≤0.3 | ≤0.05 | ≤0.04 | ≤0.3 | ≤0.008 | ≤0.3 | ≤0.08 | bal. |

The coatings AK-070, VL-023, GF-021 Express, EpoxyCoat-0263S, EpoxyCoat Zinc, EpoxyCoat-019, and EpoxyCoat Mastic were applied to a cleaned and degreased surface with a roller in one layer (EpoxyCoat-019 in two layers) and were air-dried at the room temperature (23 °C). Organosilicate coatings were applied with a roller in four layers. OS-51-03 grey and OS-56-22 grey were cured by the introduction of 0.5 wt.% of 3-aminopropyltriethoxysilane hardener agent at the room temperature, while OS-82-01 green was cured in a furnace with a heating rate of 2 °C/min and a soaking time of 3 h at a temperature of 250 °C [14].

The dry layer thickness was measured with a K6 thickness meter (Konstanta, Saint-Petersburg, Russia). A weld was made on the coated sheet side with a Motoman MH24 welding robot (Yaskawa, Kitakyushu, Japan) and an AlphaQ 552 welding machine (EWM Group, Mündersbach, Germany).

The welding was performed with 100% $CO_2$ shielding gas at a flow rate of 18 L/min and an OK Autrod 12.51 mild steel wire (ESAB Welding & Cutting) with a diameter of 1.2 mm using the following welding parameters: welding current (I)—220 ± 10 A, arc voltage (U)—22 V, travel speed ($V_{ts}$)—30 cm/min, wire feed rate ($V_{wfr}$)—7.0 m/min, and the torch was perpendicular to the substrate. The current to wire feed rate relation of 220 A to 7.0 m/min resulted in short circuiting metal transfer, during which the molten metal at the wire tip is transferred by dipping into the welding pool [28]. Thus, each metal transfer cycle consisted of an arcing period and a short circuit period.

The coatings compositions and element content are indicated in Tables 2 and 3. Coatings element content was measured using EDS method with a Mira3N scanning electron microscope (Tescan, Brno, Czech Republic) with an X-max 80 EDX detector (Oxford instruments, Abingdon, UK).

The voltage and current recorder was used to record the welding oscillograms at a 24 kHz frequency, oscillograms were recorded and processed via MATLAB, whereby the current and voltage were stored in ASCII files, and the raw data were initially smoothed using the moving-average filter:

$$y(n) = (x(n) + x(n-1) + \cdots + x(n-(w-1)))/w, \tag{1}$$

where $x(n)$ is the raw data vector, $y(n)$ the processed data vector, and $w$ the filter window size (with a value of 15). The smoothing was carried out to minimize the high frequency low amplitude oscillations on the oscillograms, caused by the proportional–integral–derivative control, performed by the welding power source.

**Table 2.** Coatings compositions and functions used in the study.

| Polymer/(Fillers + Pigments) Ratio | Binder Fillers and Pigments | Base | Thinner | Coating Class and Function | Coating |
|---|---|---|---|---|---|
| 78/22 | $SrCrO_4$ | Urea-formaldehyde resin; copolymer of methacrylic acid and methacrylic acid butyl ester | Xylene; butanol; acetone | Chemical resistant polyacrylate primer to improve adhesion of the surface to the topcoat | AK-070 |
| 47/53 | $Zn_3(PO_4)_2 \cdot 4H_2O$; $ZnCrO_4$ $4Zn(OH)_2$; $Cr_2O_3$; $Mg_3[Si_2O_5]_2(OH)_2$ | Polyvinyl butyral; formaldehyde resin | Phosphoric acid alcohol solution | Shop primer based on polyvinyl butyral | VL-023 |
| 28/72 | $Zn_3(PO_4)_2 \cdot 4H_2O$; $TiO_2$; C black; $Mg_3[Si_2O_5]_2(OH)_2$; $CaCO_3$ | PF-060 varnish; phenolic resin | Xylene | Alkyd glyphthalic primer to improve adhesion of the surface to the topcoat | GF-021 Express |
| 34/66 | $Fe_2O_3$; $Mg_3[Si_2O_5]_2(OH)_2$; $Zn_3(PO_4)_2 \cdot 4H_2O$; $ZnCrO_4 \cdot 4Zn(OH)_2$; | Bisphenol A based epoxy resin; phenol-formaldehyde resin | Xylene; butanol; acetone | Epoxy shop primer | EpoxyCoat-0263S |
| 32/68 | $Zn_3(PO_4)_2 \cdot 4H_2O$; $SrCrO_4$; $CaCO_3$; $TiO_2$; $Mg_3[Si_2O_5]_2(OH)_2$; $KAl_2(AlSi_3O_{10})(OH, F)_2$; $BaSO_4$ | Bisphenol A diglycidyl ether; castor oil ester | Xylene; Solvent | Epoxy anticorrosion primer | EpoxyCoat-019 |
| 44/56 | $Zn_3(PO_4)_2 \cdot 4H_2O$; Al powder; $TiO_2$; $Mg_3[Si_2O_5]_2(OH)_2$; $BaSO_4$ | Bisphenol A diglycidyl ether; polyoxypropylene triol triglycidyl ether | Xylene; ethyl cellosol | Two-component epoxy primer-enamel | EpoxyCoat Mastic |
| 19/81 | Zn powder; Zn flakes; $SrCrO_4$; $Mg_3[Si_2O_5]_2(OH)_2$; $BaSO_4$ | Bisphenol A Diglycidyl Ether | Xylene | Anticorrosion zinc-containing system based on epoxy resins with amine type hardener | EpoxyCoat Zinc |
| 55/45 | $Mg_3[Si_2O_5]_2(OH)_2$; $TiO_2$; $KAl_2(AlSi_3O_{10})(OH)_2$; C black | Polydimethylphenylsiloxane | Toluene | Anticorrosion, radiation-resistant, decontaminated organosilicate coating | OS-51-03 grey |
| 60/40 | $Mg_3[Si_2O_5]_2(OH)_2$; ZnO; $KAl_2(AlSi_3O_{10})(OH)_2$; $TiO_2$; $BaSO_4$; C black | Polydimethylphenylsiloxane; Polydimethylsiloxane | Toluene | Anti-icing organosilicate coating | OS-56-22 grey |
| 30/70 | $KAl_2(AlSi_3O_{10})(OH)_2$; $Cr_2O_3$ | Polymethylphenylsiloxane * | Toluene | Thermal resistant organosilicate coating | OS-82-01 green |

*—lacquer polymer is modified with organic polyester No. 315, which is synthesized from castor oil, diethylene glycol, maleic and phthalic anhydrides.

**Table 3.** Coatings element compositions.

| H * | F | Na | Cl | S | P | K | Fe | Ti | Ba wt.% | Ca | Sr | Mg | Al | Cr | Zn | Si | O | C | Coating |
|---|---|---|---|---|---|---|---|---|---|---|---|---|---|---|---|---|---|---|---|
| 5.5 | — | — | 0.1 | — | — | — | — | — | — | 0.1 | 2.5 | — | 0.2 | 1.5 | — | — | 28.7 | 61.5 | AK-070 |
| 5.8 | — | — | — | — | 1.5 | — | — | — | — | 2 | — | 3.3 | 0.2 | 0.8 | 3.8 | 3.2 | 35.6 | 43.8 | VL-023 |
| 3.4 | — | — | — | — | 0.5 | — | — | 1.9 | — | 11.2 | — | 3 | 0.2 | — | 1.4 | 3 | 45.4 | 30.1 | GF-021 Express |
| 4.4 | — | — | — | — | 0.8 | — | 14.3 | — | — | 1.5 | — | 3 | 0.2 | 0.6 | 3.3 | 2.6 | 32.8 | 36.6 | EpoxyCoat-0263S |
| 5.2 | 0.9 | — | — | 0.8 | 0.4 | — | — | 1.2 | 4.4 | 5.3 | 0.1 | 1.2 | 0.3 | 0.2 | 1.7 | 1.5 | 29.7 | 47.1 | EpoxyCoat-019 |
| 5.5 | — | — | 0.2 | 0.8 | 0.3 | — | — | 0.6 | 3.8 | 1.5 | — | 1.8 | 2.3 | — | 0.8 | 2.4 | 27.9 | 52.3 | EpoxyCoat Mastic |
| 3.8 | — | — | — | 1.2 | — | — | — | — | 6.3 | 0.4 | 1.3 | 2.1 | 0.2 | 0.8 | 24 | 1.7 | 19.2 | 39.2 | EpoxyCoat Zinc |
| 3.4 | — | — | — | — | — | 1 | 1.2 | 2.4 | — | 0.2 | — | 3.1 | 1.9 | — | — | 17.6 | 34.1 | 35.2 | OS-51-03 grey |
| 4.3 | — | — | — | 0.3 | — | 0.1 | 0.9 | 1.1 | 0.6 | 0.1 | — | 4.1 | 0.4 | — | 0.9 | 15.2 | 31.4 | 40.6 | OS-56-22 grey |
| 3.4 | — | 0.4 | — | — | — | 2.9 | 1.9 | 2.2 | — | — | — | 0.3 | 7.3 | 1.5 | — | 16.2 | 40.9 | 23 | OS-82-01 green |

*—the hydrogen amount (wt.%) is calculated based on the obtained data on the carbon content and its comparison with the coating formula composition.

To investigate the welding current deviation, a custom condition-based script was used to recognize each metal transfer cycle. The conditions were: threshold voltage, current derivative, and maximum current. Two main parameters of the welding current were measured for each cycle–peak short circuit current and mean arc current. The measured parameters were stored in the corresponding arrays. Thus, two sets of data were calculated for each coating, each set containing a series of measurements. The graphical representation of the specified welding current parameters, measured at a single metal transfer cycle, is shown in Figure 1.

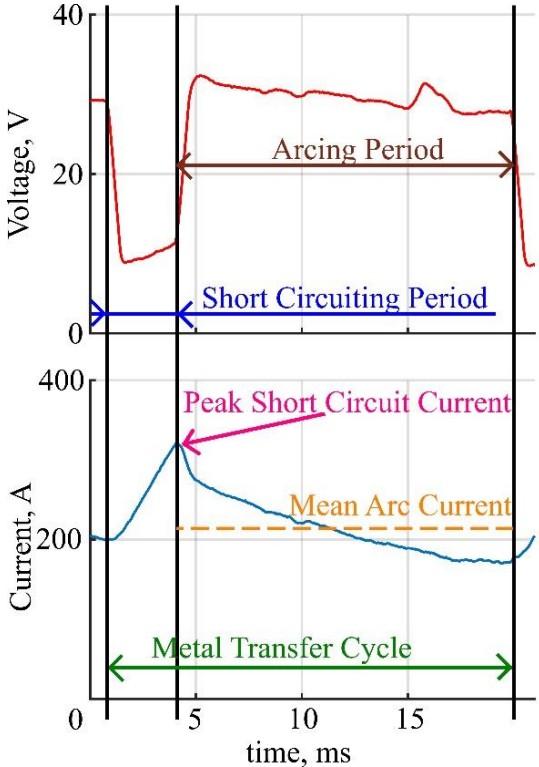

**Figure 1.** Graphical representation of measured parameters.

## 3. Results and Discussion

Coatings surface and weld seams appearance is shown in Figure 2. The coating burnout zones differs between samples. Also, spatter tracks can be spotted in samples F, H and some spatter can be spotted on the surface of samples G, I, and J. The visual quality of the seams can also be valued using Figure 2: samples A, B, C, D, F, J, and K appear to have uniform surface, while E, G, H, and I have obviously more defects on the surface.

The contact angle and the penetration form (see weld seam sections in Figure 2) considerably differs between coatings. The correlation of the contact angle and the penetration of the studied samples can be described by the Marangoni convection mode in the welding pool [29,30], when the higher surface tension of the molten metal leads to the both higher contact angle and higher penetration.

Generally, the contact angle and the penetration depth are higher when welding with coatings, especially samples I and K. However, samples F and H appear to have controversial behavior, which means that the welding pool surface tension was lower in these samples.

The weld seams and coatings parameters are given in Table 4, where $\delta$ denotes coating thickness, μm, $\varrho$ is the coating density, in g/cm$^3$, b the seam width, in mm, $b_b$ the mean width of the coating burnout zone, in mm, and H the weld bead height, in mm.

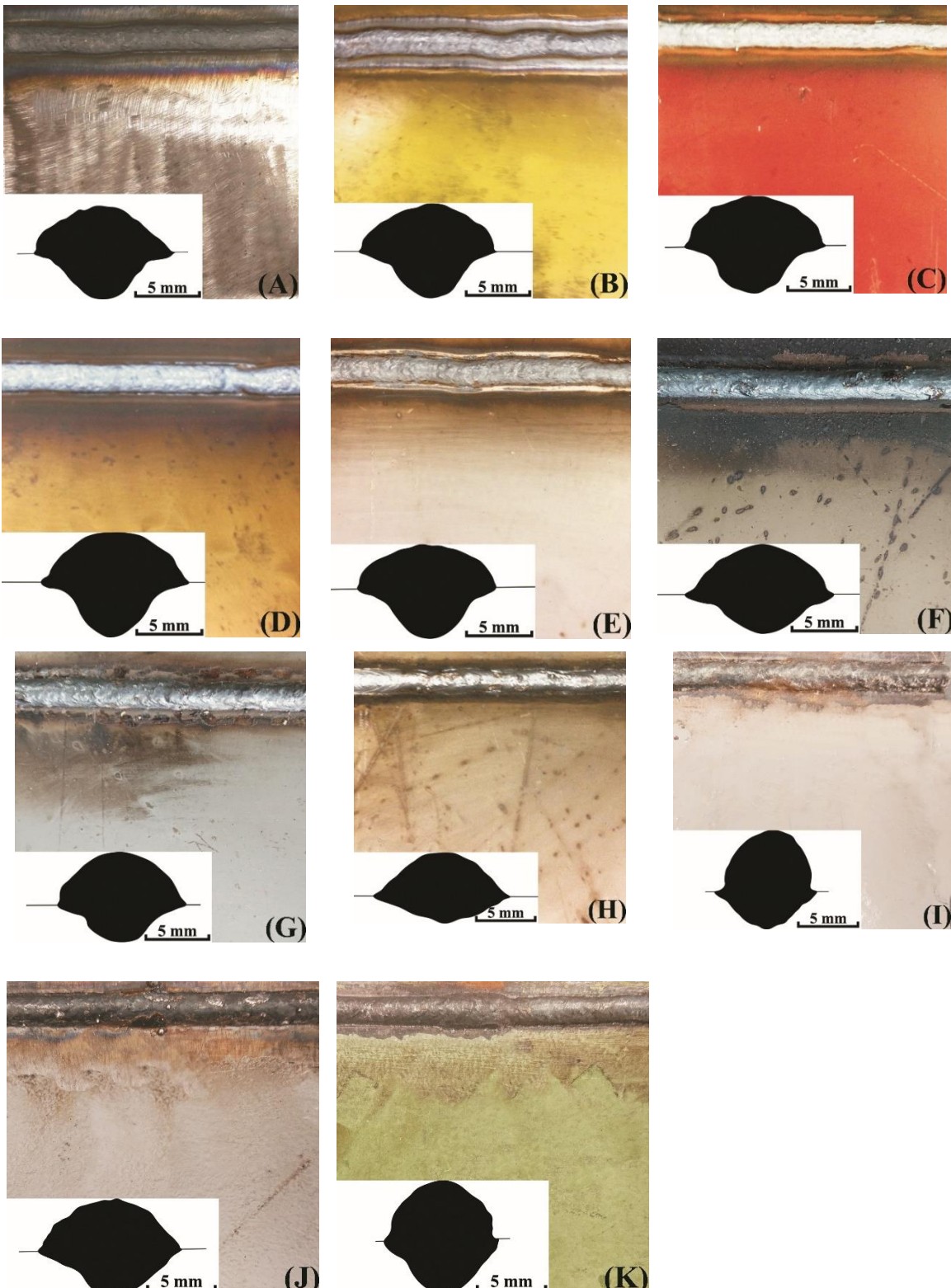

**Figure 2.** Coating surfaces and weld seam sections' profiles: (**A**) no coating; (**B**) AK-070; (**C**) VL-023; (**D**) GF-021 Express; (**E**) EpoxyCoat-0263S; (**F**) EpoxyCoat-019; (**G**) EpoxyCoat Mastic; (**H**) EpoxyCoat Zinc; (**I**) OS-51-03 grey; (**J**) OS-56-22 grey; (**K**) OS-82-01 green.

**Table 4.** The coatings and weld seams parameters.

| No. | Coating | $\delta$, μm | $\varrho$, g/cm$^3$ | $b$, mm | $b_\mathrm{b}$, mm | $H$, mm |
|-----|---------|------|------|------|------|------|
| A | no coating | 0 | — | 10.4 | — | 7.8 |
| B | AK-070 | 40 | 1.31 | 10.4 | 20.5 | 7.4 |
| C | VL-023 | 25 | 1.60 | 10.0 | 10.0 | 7.6 |
| D | GF-021 Express | 45 | 1.71 | 10.5 | 16.1 | 7.4 |
| E | EpoxyCoat-0263S | 45 | 1.90 | 10.4 | 10.4 | 7.7 |
| F | EpoxyCoat-019 | 135 | 1.66 | 9.8 | 10.1 | 6.7 |
| G | EpoxyCoat Mastic | 160 | 1.55 | 9.5 | 19.8 | 7.1 |
| H | EpoxyCoat Zinc | 100 | 2.72 | 9.6 | 12.2 | 6.0 |
| I | OS-51-03 grey | 150 | 1.49 | 9.5 | 18.9 | 7.9 |
| J | OS-56-22 grey | 185 | 1.76 | 9.9 | 41.9 | 7.0 |
| K | OS-82-01 green | 160 | 1.83 | 7.5 | 12.9 | 7.8 |

The representative parts of current and voltage waveforms captured during welding with different coatings are shown in Figure 3. These waveform parts represent a batch of subsequent metal transfer cycles, performed during each coating welding.

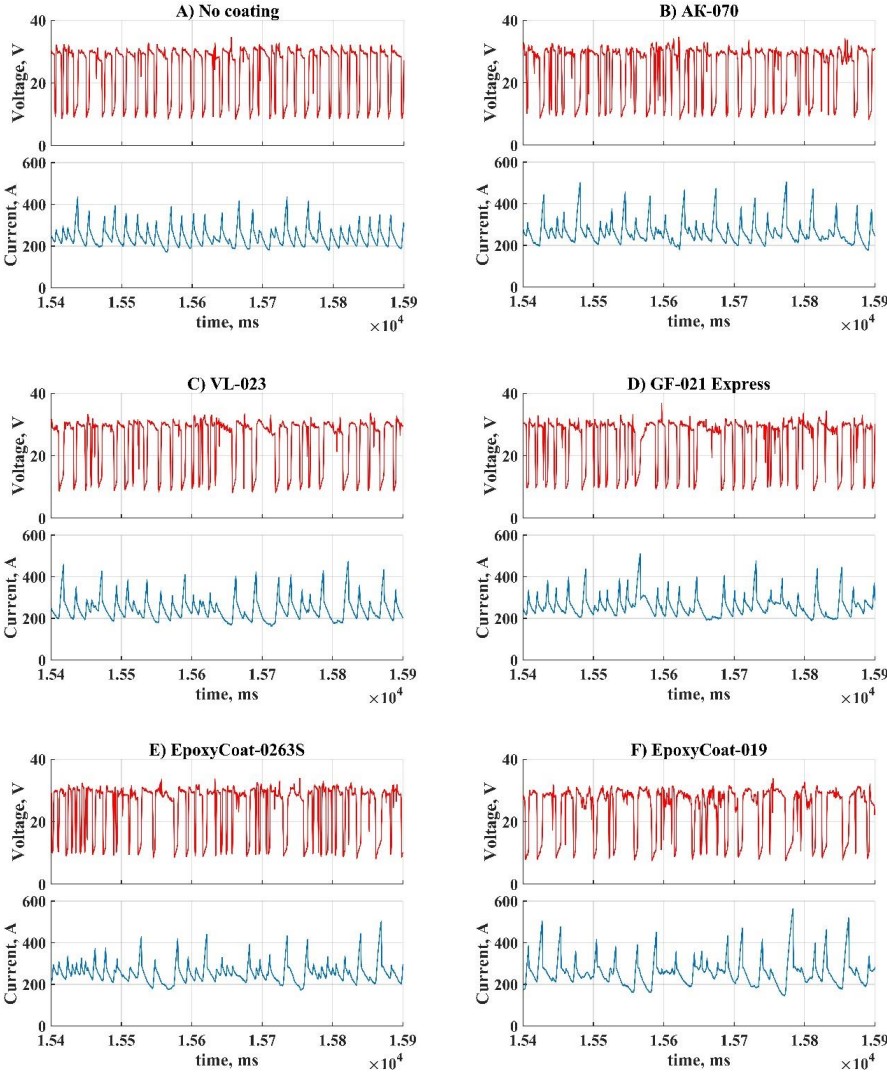

**Figure 3.** *Cont.*

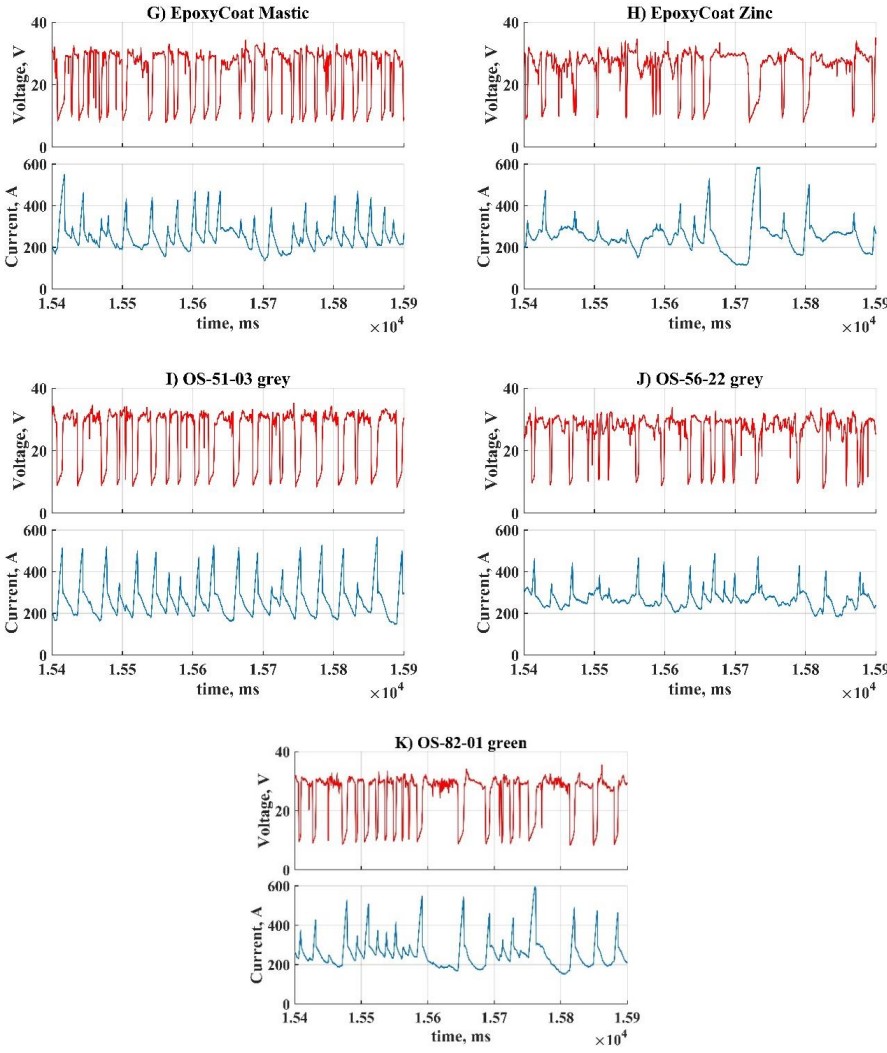

**Figure 3.** Oscillograms of the process during welding with each coating: (**A**) No coating; (**B**) AK-070; (**C**) VL-023; (**D**) GF-021 Express; (**E**) EpoxyCoat-0263S; (**F**) EpoxyCoat-019; (**G**) EpoxyCoat Mastic; (**H**) EpoxyCoat Zinc; (**I**) OS-51-03 grey; (**J**) OS-56-22 grey; (**K**) OS-82-01 green.

During welding without coating (Figure 3A) metal transfer cycles are repeated mostly uniformly: the short circuit current does not rise higher than 400 A, the arc burns without significant distortion of current. During welding with coatings (Figure 3B–F,H,J) significant distortions of the arc current can be observed, though some coatings (Figure 3G,I,K) almost did not affect the arc current. Each coating influenced the short circuit current so that the peak arc current deviation dramatically increased.

The welding current stability was studied using the variation coefficients of the peak short circuit current and the mean arc current. The variation coefficient is determined as the ratio of the standard deviation and mean value for a set of data [31]. In this study, the variation coefficient was calculated for two sets of data for each coating: the first set measured for each metal transfer cycle peak short circuit current and the second set measured for each metal transfer cycle mean arc current. The coefficients' values are represented in Figure 4.

The variation coefficients for both welding current parameters are low [31] and differ insignificantly between coatings, which means that the end of the short-circuiting period is repeated constantly, and the arc burns uniformly during welding with each coating. However, different welding current deviation is observed in the oscillograms between coatings. To investigate this deviation, the histogram profiles of two main parameters distributions were plotted for each coating: the studied parameters were peak short circuit current and mean arc current. Their distributions were calculated from arrays,

where the parameters' values calculated for each transfer cycle were stored. Histograms are presented in Figures 5 and 6. It should be noted that these histograms were calculated for the parameters' distributions, that were obtained during processing of the whole oscillograms of each coating welding.

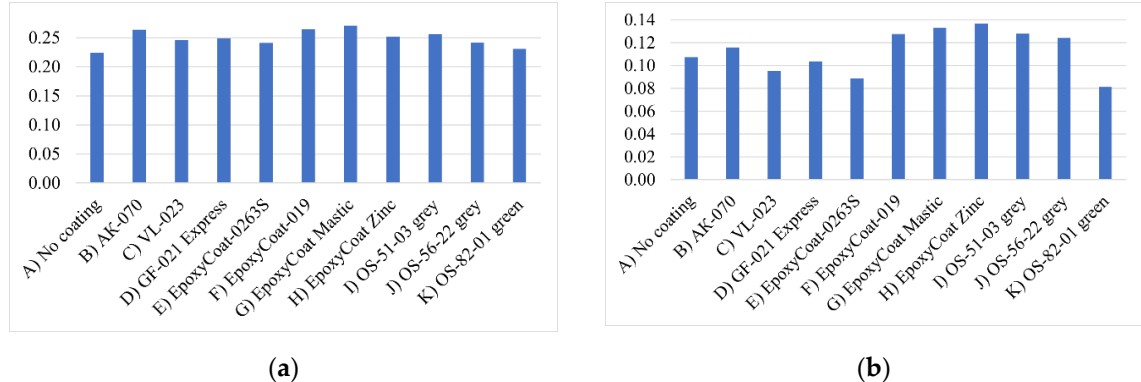

(**a**)                                                                                       (**b**)

**Figure 4.** The variation coefficient of: (**a**) peak short circuit current; (**b**) mean arc current.

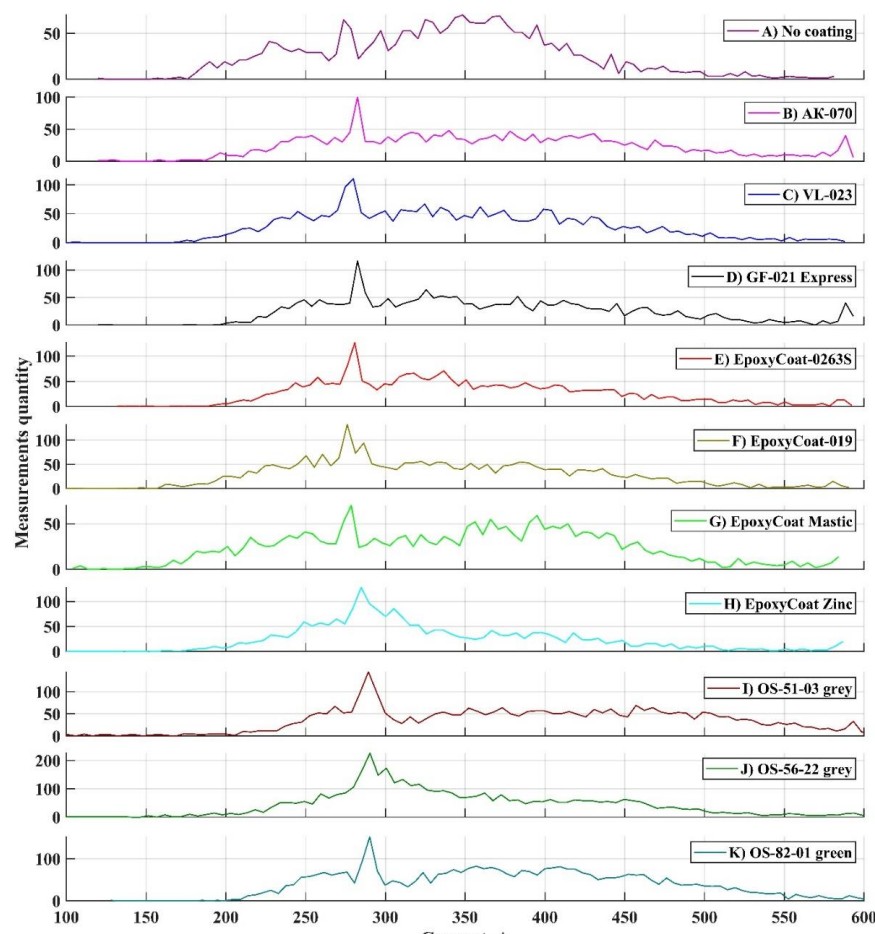

**Figure 5.** Profiles of peak short circuit current histograms: (**A**) No coating; (**B**) AK-070; (**C**) VL-023; (**D**) GF-021 Express; (**E**) EpoxyCoat-0263S; (**F**) EpoxyCoat-019; (**G**) EpoxyCoat Mastic; (**H**) EpoxyCoat Zinc; (**I**) OS-51-03 grey; (**J**) OS-56-22 grey; (**K**) OS-82-01 green.

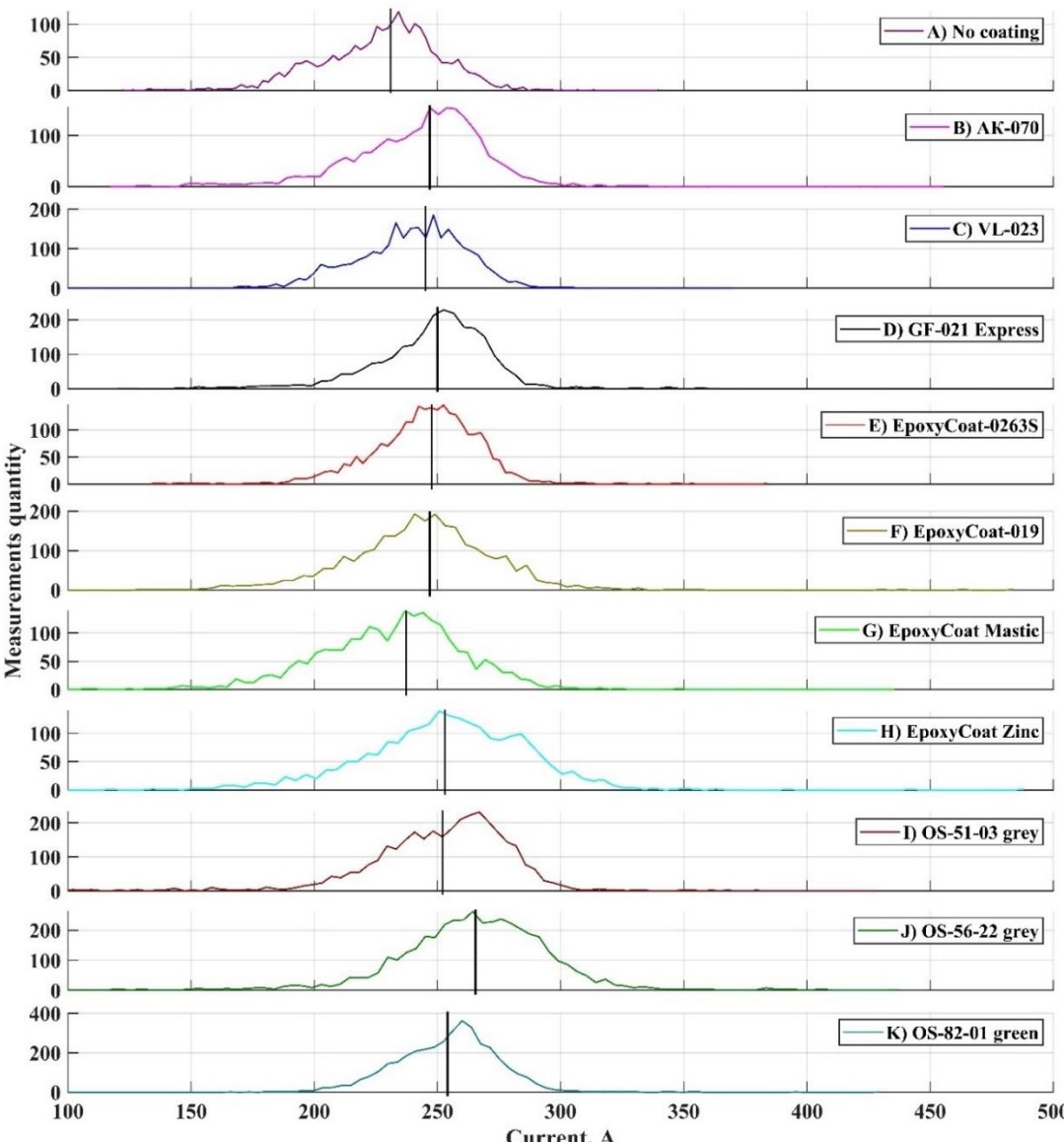

**Figure 6.** Profiles of mean arc current histograms: (**A**) No coating; (**B**) AK-070; (**C**) VL-023; (**D**) GF-021 Express; (**E**) EpoxyCoat-0263S; (**F**) EpoxyCoat-019; (**G**) EpoxyCoat Mastic; (**H**) EpoxyCoat Zinc; (**I**) OS-51-03 grey; (**J**) OS-56-22 grey; (**K**) OS-82-01 green.

Peak short circuit current distribution (Figure 5) in case of welding without coating is mostly uniform and fits within the range of 150–450 A. Each other distribution has significant increase in the upper limit of this range: peak arc current is often higher than 450 A, the right "tail" of the distribution is smoothed to the higher values of current.

During short circuiting period of the metal transfer cycle [28], the molten metal is transferred from the wire tip to the welding pool mainly under the surface tension force [32], the gravity and the pinch effect [33]. The surface tension of the welding pool and the bridge between the pool and the tip is influenced by the pool chemical composition [29]. When the surface active element, such as oxygen or sulfur, exceeds a certain value in the welding pool [34,35], the surface tension temperature coefficient changes from a negative to a positive value, which means that the surface tension increases with temperature increasing. Compared to the welding without coating, an increased surface tension leads to an increase in the opposing pinch force value, which depends on the current, so the peak short circuit current rises [32].

Each distribution, as presented in Figure 5, has a noticeable peak between 250 and 300 A. Most of the short circuit terminations occurred at a current, marked by this distribution peak. It can be noted that in case of organosilicate coatings, this peak of distribution is situated at a considerably higher current. This behavior correlates with the weld seam section profiles of welds with organosilicate coatings: these coatings provide higher contact angle (see contact angles in Figure 2), which corresponds to higher surface tension. This fact also confirms relation of peak short circuit current and surface tension.

Mean arc current histograms (Figure 6) appear as normal distributions, yet its' peaks deviate in a peculiar way. In case of welding without any coating, the mean arc current is at its lowest value. Moreover, coatings influenced the mean arc current differently: mean values of distributions, that marked by vertical lines in Figure 6, differ between coatings. To investigate this behavior of the arc current, each element substance amount in each coating was calculated using formula:

$$\text{Amount} = b \times b \times \delta \times \varrho \times \text{wt.\%}/M, \tag{2}$$

where wt.% is the obtained element content (Table 3) and $M$ the element's molar mass, g/mol. The area of arc burning is assessed as $b \times b$ in formula 2. Then the mean arc current was compared to each element amount. The correlation was found between arc current and oxygen amount. The relation is represented in Figure 7.

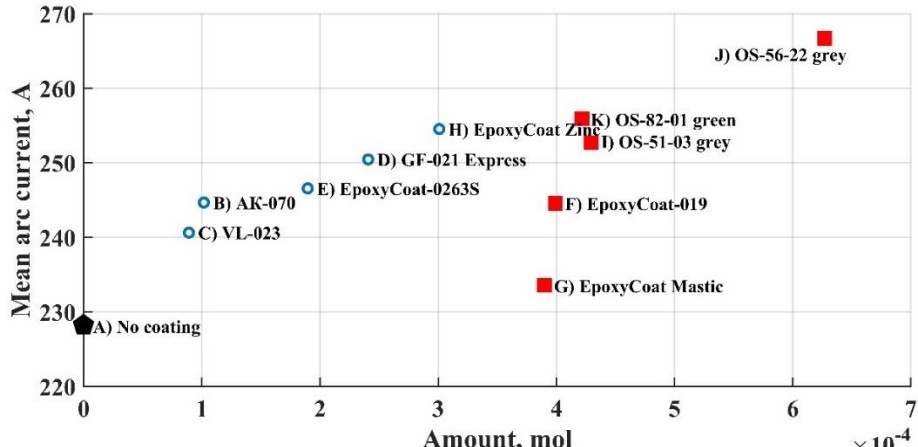

**Figure 7.** The mean arc current vs. oxygen amount.

The higher the oxygen amount in the coating, the higher is the mean arc current. This relation tends to be linear, but there are two groups of coatings in the graph (Figure 7): groups A (blue circles) and B (red squares). Such a division in two groups can be explained by difference in thickness: in group A coatings have thickness equal or less than 100 μm, and in group B they are significantly thicker. Thickness could have influenced the arc current in a complex way by the drastic difference in the amount of any substances produced during coatings breakdown, but the main relation between the oxygen amount and the arc current can be explained through the carbon monoxide formation during welding.

Two basic elements that form each coating are carbon and oxygen (see Table 3). Since they form complex compositions in the coatings–the element biding energy in these compositions tends to be low. Weakness of bonds between carbon or oxygen and other elements can lead to the release of atomic carbon and oxygen during coating breakdown in high arc temperature, which can lead to the formation of carbon dioxide ($CO_2$) or carbon monoxide (CO) [36]:

$$2C + O_2 = 2CO, \tag{3}$$

$$C + O_2 = CO_2, \tag{4}$$

During gas metal arc welding in the $CO_2$ shielding gas, $CO_2$ dissociates to CO and $O_2$ in the arc since its dissociation degree significantly rises at 2000 K [37]. CO reacts with the plasma in the arc [38,39], and since CO ionization energy is higher than that of $CO_2$, the arc voltage increases and hence the arc current.

Since the chemical equilibrium of reactions (3) and (4) in a $CO_2$ shielding gas is shifted towards the formation of CO and the temperature near the cathode spot around 3300 K boosts the dissociation of $CO_2$ [37], CO accounts for the largest share of the gas formed. Since the amount of released oxygen and formed $CO_2$ is equal, accounting that there is always more than enough carbon, the amount of CO, generated from the coating during welding, can be calculated using formula:

$$\text{Amount} = b \times l \times \delta \times \varrho \times \text{wt.\%}/M, \tag{5}$$

where $l$ is the seam length, in mm. Using the molar volume of ideal gases, the CO generated volume (after it leaves the arc) can be calculated for each coating. These volumes are indicated in Table 5. Since the welding time is 30 s, the CO generation rate can also be easily calculated.

**Table 5.** Volume of the generated CO from the coatings' breakdown products.

| Coating | CO Volume, mL | CO Generation Rate, mL/min |
|---|---|---|
| AK-070 | 99 | 199 |
| VL-023 | 52 | 104 |
| GF-021 Express | 71 | 141 |
| EpoxyCoat-0263S | 95 | 191 |
| EpoxyCoat-019 | 306 | 612 |
| EpoxyCoat Mastic | 366 | 731 |
| EpoxyCoat Zinc | 298 | 596 |
| OS-51-03 grey | 217 | 434 |
| OS-56-22 grey | 383 | 765 |
| OS-82-01 green | 150 | 300 |

Ojima [40] evaluated the CO generation rate from $CO_2$ during GMAW at different current and different shielding gas flow rates. Based on the data in [40], the usual CO generation rate during GMAW is about 500 mL/min. The CO generation rates, indicated in Table 5, are comparable to the mean value, obtained without any coating. This explains the correlation between mean arc current and oxygen amount in the coatings.

## 4. Conclusions

Experimental studies of the welding current waveforms were performed with epoxy, alkyd, polyacrylate, polyvinyl butyral primers, epoxy zinc filled, vinyl chloride, vinyl isobutyl, and organic-silicate coatings. The following conclusions can be drawn from the present research:

1. Short circuit current and arc current significantly deviate when welding with coatings due to the influence of coatings breakdown products on the welding pool and on the arc.
2. The peak short circuit current increases and deviates more when welding with coatings due to the increase in surface tension, caused by the presence of oxygen in the welding pool.
3. CO, generated during the breakdown of the coating, significantly increases the mean arc current. The higher the oxygen content in the coating, the greater the influence of the generated CO.

The relations between the oscillograms and the physical processes that take place during welding with coatings discovered in the present study will be used in future work, dedicated to the development of the welding job for defectless welding with different coatings.

**Author Contributions:** Conceptualization, L.Z. and D.K.; data curation, I.M.; funding acquisition, O.P.; investigation, L.Z. and D.K.; methodology, L.Z. and D.K.; project administration, O.P.; software, I.M.; validation,

I.M.; writing—original draft, L.Z. and D.K.; writing—review and editing, O.P. All authors have read and agreed to the published version of the manuscript.

**Funding:** This research work was supported by the Academic Excellence Project 5-100 proposed by Peter the Great St. Petersburg Polytechnic University.

**Conflicts of Interest:** The authors declare no conflict of interest.

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
