# Peer review of "The Coatings Breakdown Products Influence on the Gas Metal Arc Welding Parameters"

_coatings, doi:10.3390/coatings10111061_

Round 1

Reviewer 1 Report

Dear Authors,

The article concerns the experimental verification of the effect of breakdown products of few types of coatings deposited on mild steel on GMAW parameters. The subject of the reviewed submission is important from a practical and scientific point of view, and I’m convinced that manuscript should be published. Please consider making the proposed changes:

The term "destruction products" is used in literature several dozen times less frequently than "breakdown products" (and mainly by authors who are not native speakers).

Throughout the text and tables, please replace commas with dots.

Chapter 2: the description of the welding procedure needs to be supplemented so that the reader can repeat the experiment:

What grade of steel was used? Please provide designation and chemical composition. How are coatings applied to the steel surface? What welding machine was used?

Line 113: change: "job" to "parameters".

According to the Mdpi guidelines, the articles should include the manufacturers of the devices used.

Insert spaces in Table 1.

Line 132: What is smoothing used for? Elimination of measurement errors?

Figure 1: the figure looks unprofessional: the charts are small and the font is large. I propose to remove the legends from "Voltage" and "Current". Similarly in Figure 3.

Line 161: Correct the g/cm3 unit notation.

"Spattering", "spatters" - should be: "spatter"

Table 3: How can you explain the negative delta m value for AK-017?

The analysis of the results based on the plots is difficult. Is it possible to carry out a quantitative analysis using the coefficients describing the stability of the welding arc according to: Wang, J., Sun, Q., Zhang, T., Tao, X., Jin, P., & Feng, J. (2019). Arc stability indexes evaluation of ultrasonic wave-assisted underwater FCAW using electrical signal analysis. The International Journal of Advanced Manufacturing Technology103(5-8), 2593-2608?  

Line 241: change: “half a minute” to “30 s”.

Use the following vocabulary consistently: "arc voltage" and "welding current". Similarly: "mean" and "average".

Consistently use either the symbols of chemical compounds (CO) or their names (carbon monoxide).

Authors contributions: The MDPI guidelines only lists the initials of the authors.

References: please format them carefully: eg [6]: no place of the conference, [33]: there should be a different bibliographic description of the journal.

Author Response

Dear Reviewer,

thank you for your valuable review. Here is a detailed response:

Reviewer #1: The article concerns the experimental verification of the effect of breakdown products of few types of coatings deposited on mild steel on GMAW parameters. The subject of the reviewed submission is important from a practical and scientific point of view, and I’m convinced that manuscript should be published. Please consider making the proposed changes:

Reviewer #1 comment (1). The term "destruction products" is used in literature several dozen times less frequently than "breakdown products" (and mainly by authors who are not native speakers).

Response (1): Every phrase “destruction product” is replaced with “breakdown product” in the manuscript.

Reviewer #1 comment (2). Throughout the text and tables, please replace commas with dots.

Response (2): Throughout the text and tables, commas are replaced with dots in the manuscript.

Reviewer #1 comment (3). Chapter 2: the description of the welding procedure needs to be supplemented so that the reader can repeat the experiment: What grade of steel was used? Please provide designation and chemical composition. How are coatings applied to the steel surface? What welding machine was used?.

Response (3): The table with the chemical composition of the used steel is added to the manuscript. The coatings application procedure is clarified in the manuscript. The welding equipment is indicated along with its manufacturer.

Reviewer #1 comment (4). Line 113: change: "job" to "parameters".

Response (4): The word “job” is replaced with the word “parameters”.

Reviewer #1 comment (5). 5.     According to the Mdpi guidelines, the articles should include the manufacturers of the devices used.

Response (5): The used devices manufacturers are indicated using parentheses in the second chapter of the manuscript.

Reviewer #1 comment (6). Insert spaces in Table 1.

Response (6): Spaces are inserted in Table 1.

Reviewer #1 comment (7). Line 132: What is smoothing used for? Elimination of measurement errors?.

Response (7): The smoothing was carried out to minimize the high frequency low amplitude oscillations on the oscillograms, caused by the proportional–integral–derivative control, performed by the welding power source. The purpose of the smoothing is now clarified in the manuscript.

Reviewer #1 comment (8). Figure 1: the figure looks unprofessional: the charts are small and the font is large. I propose to remove the legends from "Voltage" and "Current". Similarly in Figure 3..

Response (8): The font size in the figure 1 was corrected, the legend was excluded from figures 1 and 3.

Reviewer #1 comment (9). Line 161: Correct the g/cm3 unit notation.

Response (9): g/cm3 unit notation is corrected.

Reviewer #1 comment (10). 10. "Spattering", "spatters" - should be: "spatter".

Response (10): The word “spatter” is now corrected throughout the manuscript.

Reviewer #1 comment (11). Table 3: How can you explain the negative delta m value for AK-017?

Response (11): The delta m value cannot be negative – this is the result of miscalculation. Since the result of approximate mass loss measurement is not used in the discussion, the mass loss column is excluded from table 3 (table 4 in the revised version of the manuscript).

Reviewer #1 comment (12). The analysis of the results based on the plots is difficult. Is it possible to carry out a quantitative analysis using the coefficients describing the stability of the welding arc according to: Wang, J., Sun, Q., Zhang, T., Tao, X., Jin, P., & Feng, J. (2019). Arc stability indexes evaluation of ultrasonic wave-assisted underwater FCAW using electrical signal analysis. The International Journal of Advanced Manufacturing Technology, 103(5-8), 2593-2608?.

Response (12): A quantitative analysis of the results based on the plots using the coefficients describing the stability of the welding arc was carried out according to the suggested article. The results are added and discussed in the manuscript.

Reviewer #1 comment (13). Line 241: change: “half a minute” to “30 s”.

Response (13): The phrase “half a minute” is replaced with “30 s”.

Reviewer #1 comment (14). Use the following vocabulary consistently: "arc voltage" and "welding current". Similarly: "mean" and "average".

Response (14): There is practical difference between welding current and arc current in the manuscript: welding current is the the current, that goes through the welding wire, through the welding pool, through the workpiece, etc., at any moment of the welding. Arc current is the welding current that goes through the arc only when it burns. Since the arc does not burn when the short circuit occurs, the arc current is separated from the short circuit current. This is now clarified in the second chapter of the manuscript. The word “average” is now left only in one case, in the phrase “at average”. The word “mean” is chosen in the manuscript since it describes arithmetical mean values, that were used in the study.

Reviewer #1 comment (15). Consistently use either the symbols of chemical compounds (CO) or their names (carbon monoxide).

Response (15): Symbols of chemical compounds are now consistently used.

Reviewer #1 comment (16). Authors contributions: The MDPI guidelines only lists the initials of the authors.

Response (16): Authors contributions section is corrected according to the guidelines.

Reviewer #1 comment (17). 17. References: please format them carefully: eg [6]: no place of the conference, [33]: there should be a different bibliographic description of the journal.

Response (17): References 6 and 33 were corrected according to the guideline for the bibliographic description of the conference paper – the places of the conferences are added.

Reviewer 2 Report

The authors presented an interesting study on the influence of the coating destruction products on the gas metal arc welding process parameters. The paper is well structured by presenting successively the introduction followed by the experimental approach along with the results and conclusion. The results are deeply discussed, and valued conclusions are made. However, there are some minor comments, addressing which can improve the work presentation:

  1. The state-of-the-art lacks reviewing new and recent articles. You may briefly point out the coatings in other types of welding such as wet welding: The abrasive wear resistance of coatings manufactured on high-strength low-alloy (HSLA) offshore steel in wet welding conditions. (2020). Coatings10(3), 219.
  2. The correlation of the contact angle and the weld penetration is well explained. I see a lack of discussion on how the welding pool temperature is correlated to welding pool fluid dynamics (movements) and dimensions. How considerable is the recoil pressure due to the evaporated gas? Did you find a temperature at which the recoil pressure is considerable so it overcomes the surface tension and significantly changes the welding pool geometry? Please refer to the following article and briefly point out how these forces change the welding pool geometry: A review on melt-pool characteristics in laser welding of metals. (2018). Advances in Materials Science and Engineering2018.
  3. Valuable conclusions are made; however, the conclusion section seems incomplete. After presenting point by point conclusions, please add a short paragraph on how this information can be useful for future work.
  4. A proper improvement in English is recommended.

Author Response

Dear Reviewer,

thank you for your valuable review. Here is a detailed response:

Reviewer #2: The authors presented an interesting study on the influence of the coating destruction products on the gas metal arc welding process parameters. The paper is well structured by presenting successively the introduction followed by the experimental approach along with the results and conclusion. The results are deeply discussed, and valued conclusions are made. However, there are some minor comments, addressing which can improve the work presentation:

Reviewer #2 comment (1). The state-of-the-art lacks reviewing new and recent articles. You may briefly point out the coatings in other types of welding such as wet welding: The abrasive wear resistance of coatings manufactured on high-strength low-alloy (HSLA) offshore steel in wet welding conditions. (2020). Coatings, 10(3), 219.

Response (1): More recent articles are reviewed in the introduction part of the manuscript, including the suggested one.

Reviewer #2 comment (2). The correlation of the contact angle and the weld penetration is well explained. I see a lack of discussion on how the welding pool temperature is correlated to welding pool fluid dynamics (movements) and dimensions. How considerable is the recoil pressure due to the evaporated gas? Did you find a temperature at which the recoil pressure is considerable so it overcomes the surface tension and significantly changes the welding pool geometry? Please refer to the following article and briefly point out how these forces change the welding pool geometry: A review on melt-pool characteristics in laser welding of metals. (2018). Advances in Materials Science and Engineering, 2018.

Response (2): According to :DebRoy, T.; David, S.A. Physical processes in fusion welding. Rev. Mod. Phys. 1995, 67, 85–112, doi:10.1103/RevModPhys.67.85., “significant vaporization from the weld pool takes place when very high power-density energy sources such as lasers and electron beams are used for welding. In many cases, the escaping vapor exerts a large recoil force, and, as a result, the molten metal is expelled from the cavity.” Since the arc welding was used in the present study, the recoil pressure was not considered due to the relatively low energy density of the welding arc. The arc energy input was similar between coatings because the same welding job was used, which is why the welding pool temperature was not measured or calculated in this study. However, the suggested article contains a complete description on the other forces, that were discussed in the manuscript. The description of Marangoni forces in the manuscript is now referred to the suggested article.

Reviewer #2 comment (3). Valuable conclusions are made; however, the conclusion section seems incomplete. After presenting point by point conclusions, please add a short paragraph on how this information can be useful for future work.

Response (3): A short paragraph on how conclusions can be useful for the future work is now added in the fourth chapter of the manuscript.

Reviewer #2 comment (4). A proper improvement in English is recommended.

Response (4): An improvement in English is made throughout the whole manuscript.

Round 2

Reviewer 1 Report

Dear Authors,
thank you very much for the answers and for completing the manuscript according to my suggestions. Your answers are extremely professional. Congratulations on taking your review process seriously and on a very good article! During the author's correction, pay attention to replacing commas with dots in Figure 4, besides, chart titles can be removed because they duplicate the information from the figure caption. Write "table 1" and "table 5" in capital letters. Best regards